# D-O-A based organic phosphors for both aggregation-induced electrophosphorescence and host-free sensitization

Lulin Xu[1,6], Yuhang Mo[1,6], Ning Su[1], Changshen Shi[2] ✉, Ning Sun[2], Yuewei Zhang[3] ✉, Lian Duan[3], Zheng-Hong Lu[2,4] & Junqiao Ding[1,5] ✉

Pure organic phosphors capable of room-temperature phosphorescence show a great potential in organic light-emitting diodes, while it is limited by the big challenge to realize efficient electroluminescence under electric excitation. Herein, we develop a class of organic phosphors based on acridine as the electron donor, triazine as the electron acceptor and oxygen as the bridge between them. Benefitting from the characteristic donor-oxygen-acceptor geometry, these compounds are found to behave an exciting aggregation-induced organic room-temperature electrophosphorescence, and achieve a record-high external quantum efficiency of 15.8% for non-doped devices. Furthermore, they can sensitize multi-resonant emitters in the absence of any additional wide bandgap host, leading to an effective narrowband emission with a peak external quantum efficiency of 26.4% and a small full-width at half maximum of 26 nm. The results clearly indicate that donor-oxygen-acceptor geometry is a promising strategy to design organic phosphors suitable for organic light-emitting diodes.

Compared to metal-containing ones[1], pure organic phosphors capable of room-temperature phosphorescence (RTP) have drawn much attention recently because of their low cost, good biocompatibility and ease of tailorability[2–4]. Aiming at organic RTP, not only the intersystem crossing (ISC) process is required to be facilitated through lone-pair electron incorporation[5], heavy-atom effect[6], hyperfine coupling[7], energy-gap narrowing[8] and molecular aggregation[9], but also the non-radiative decay and environmental quenching should be minimized via crystallization[10,11], polymerization[12,13], host-guest complexation[14], matrix rigidification[15], crosslinking[16] and clusterization[17]. Based on these design principles[18], many interesting organic RTP systems have been demonstrated up to now, whose photoluminescence (PL) under photo excitation is deeply elucidated and thus widely applied in functional sensors, information encryption and bioimaging[19–21].

However, the promising electroluminescence (EL) under electric excitation is seldom explored for organic RTP emitters[22–31], although they have a great potential in organic light-emitting diodes (OLEDs) to harvest both singlet and triplet excitons so as to achieve a theoretical 100% internal quantum efficiency (Fig. 1a). In this case, there exists an unwanted dilemma. That is, most of the above-developed conditions to boost bright RTP emission in the PL process are difficult to be satisfied in the EL process[24]. Moreover, it still remains a black box on how to design high-performance organic RTP emitters suitable for OLEDs at present. Therefore, novel universal strategies are extremely desirable for the design of organic phosphors showing efficient organic room-temperature electrophosphorescence.

Previously, we reported a conversion from fluorescence to RTP after an oxygen atom (O) was inserted into a donor-acceptor (D-A)

[1]School of Chemical Science and Technology, Yunnan University, 650091 Kunming, People's Republic of China. [2]School of Physics and Astronomy, Yunnan University, 650091 Kunming, People's Republic of China. [3]Key Lab of Organic Optoelectronics and Molecular Engineering of Ministry of Education, Department of Chemistry, Tsinghua University, 100084 Beijing, People's Republic of China. [4]Department of Materials Science and Engineering, University of Toronto, Toronto, ON, Canada. [5]Southwest United Graduate School, 650092 Kunming, People's Republic of China. [6]These authors contributed equally: Lulin Xu, Yuhang Mo. ✉e-mail: csshi@ynu.edu.cn; 15764322641@163.com; dingjunqiao@ynu.edu.cn

(a) Previous research about organic RTP focusing on PL rather than EL

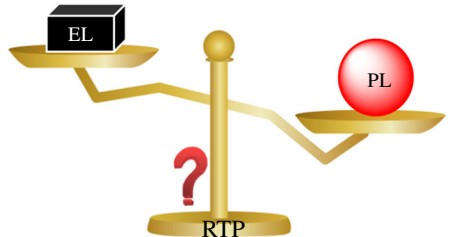

(b) Molecular design rule of organic RTP emitters suitable for EL (this work)

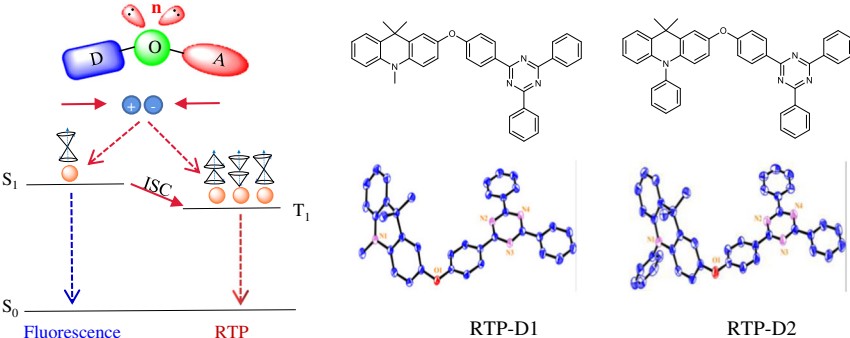

**Fig. 1 | Driving force to develop organic phosphors for OLEDs. a** Schematic diagram of previous research about organic RTP focusing on PL rather than EL; **b** Schematic diagram of molecular design rule of organic RTP emitters suitable for EL together with molecular and crystal structures of developed organic phosphors.

conjugated polymer[32]. Meanwhile, due to the good electroactivity of the resultant donor-oxygen-acceptor (D-O-A)-based polymer, the corresponding doped OLEDs achieved a maximum external quantum efficiency (EQE) of 9.7%. However, it still remains a doubt whether the D-O-A structure could realize efficient organic room-temperature electrophosphorescence at a small molecular level[33], when given the difference between polymers and small molecules including conjugation length, intermolecular interactions, rotation possibility around the C–O single bond etc.

With this idea in mind, a class of metal-free organic phosphors named RTP-D1 and RTP-D2 have been developed by using acridine as D, triazine as A and O as the bridge between D and A (Fig. 1b). Owing to the involvement of the n-orbital of the oxygen bridge, the characteristic D-O-A geometry is expected to favor the effective ISC from singlet to triplet excitons and subsequent phosphorescence generation from triplet excitons. As a consequence, both the RTP-D1 and RTP-D2 neat films show an exciting aggregation-induced organic room-temperature electrophosphorescence by themselves, revealing a record-high EQE of 15.8% (45.8 cd/A, 50.4 lm/W) and Commission Internationale de l'Eclairage (CIE) coordinates of (0.16, 0.55) for non-doped OLEDs. To our knowledge, the performance is the highest ever reported for organic RTP emitters, representing an important progress towards their challenging EL application rather than PL. Most importantly, in the absence of any additional wide bandgap host, they can also be used to sensitize multiple resonance emitter to achieve efficient narrowband emission with a peak EQE of 26.4% (54.0 cd/A, 63.0 lm/W) and a small full-width at half maximum (FWHM) of 26 nm.

## Results

### Synthesis and characterization

RTP-D1 and RTP-D2 only differ from each other in the substituent of the acridine donor (Supplementary Fig. 1). The same nucleophilic aromatic substitution reaction was carried out between 2-(4-fluorophenyl)−4,6-diphenyl-1,3,5-triazine and hydroxyl intermediates to afford RTP-D1 and RTP-D2 in the yield of 62% and 60%, respectively.

Their molecular structures were fully characterized by using $^1H$ and $^{13}C$ NMR, mass spectra, elemental analysis and single crystals (Supplementary Figs. 2–6 and Supplementary Table 1). Also, they are thermally stable, showing a decomposition temperature ($T_d$, corresponding to a 5% weight loss) of 338–351 °C and a glass transition temperature ($T_g$) of 83–97 °C (Supplementary Fig. 7).

Cyclic voltammetry (CV) was measured to study the electrochemical properties of RTP-D1 and RTP-D2. As one can see, both compounds display quasi-reversible oxidation and reduction signals during the anodic and cathodic sweeping (Supplementary Fig. 8). The observed good electroactivity is expected to favor the charge transporting when they are used for OLEDs. Based on the electrochemical data and ferrocene/ferrocenium (Fc/Fc$^+$) as the standard (−4.8 eV under vacuum), the highest occupied and lowest unoccupied molecular orbital (HOMO/LUMO) levels are determined to be −5.18/−2.74 eV for RTP-D1 and −5.26/−2.76 eV for RTP-D2 (Table 1). Noticeably, RTP-D2 exhibits a close LUMO but a slightly deeper HOMO than RTP-D1, which may be attributed to the different substituents on acridine. As for RTP-D2, the conjugated phenyl substituent on acridine could weaken the electron cloud density of acridine and thus reducing its electron-donating capability, leading to a reduced HOMO level[34].

### Photophysical properties

Figure 2 depicts the UV–Vis absorption in dichloromethane, PL spectra in cyclohexane without $O_2$, PL and phosphorescence spectra in neat films for RTP-D1 and RTP-D2. Since their HOMO and LUMO are distributed on acridine and triazine (Supplementary Fig. 9), respectively, there exists a weak charge transfer (CT) from acridine to triazine in spite of the O insertion between them. According to our previous work[32], the related CT absorption can be reasonably ascribed to the tail ranging from 350 to 400 nm. Moreover, the intense band in the range of 250-350 nm originates from the n−π* and/or π−π* transitions of the individual acridine and triazine fragments. In agreement with the above electrochemical properties, additionally, the PL film of RTP-D2 is sensitive to $O_2$ obviously (Supplementary Fig. 10), and blue-shifted by

**Table 1 | Summary of the photophysical, electrochemical and thermal properties for RTP-D1 and RTP-D2**

| Emitter | $\lambda_{abs}$[a] [nm] | $\lambda_{PL}$[b] [nm] | $\lambda_P$[c] [nm] | $\Phi_{PL}$[d] [%] | $\tau_F/\tau_P$[e] [ns] | HOMO/LUMO[f] [eV] | $T_d$[g] [°C] | $T_g$[h] [°C] |
|---|---|---|---|---|---|---|---|---|
| RTP-D1 | 277 | 504 (87) | 511 | 54.9 | 16.8/488.8 | −5.18/−2.74 | 338 | 83 |
| RTP-D2 | 280 | 497 (91) | 505 | 77.5 | 12.6/642.7 | −5.26/−2.76 | 351 | 97 |

[a]Absorption measured in $10^{-5}$ mol L$^{-1}$ dichloromethane solution.
[b]PL measured in neat film and the related FWHM values are listed in the parentheses.
[c]RTP measured in neat film under a 0.1 ms delay between the pulsed excitation and the emission collection.
[d]The $\Phi$PLs measured in film under $N_2$ using an integrating sphere.
[e]Fluorescence and RTP lifetimes estimated from the transient PL spectra in neat film.
[f]HOMO and LUMO levels determined by CV.
[g]Decomposition temperature corresponding to a 5% weight loss.
[h]Glass transition temperature taken from the middle value between break points.

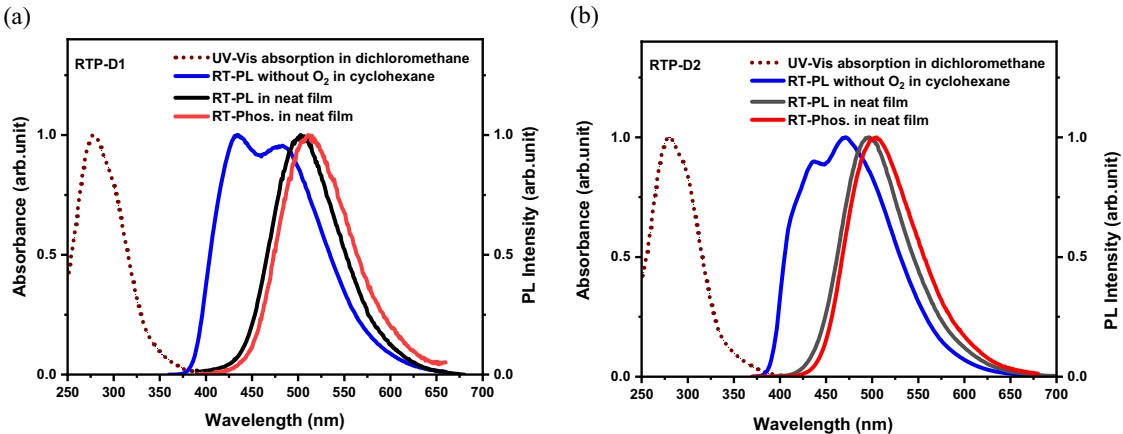

**Fig. 2 | The photophysical properties. a**, **b** UV–Vis absorption spectra in dichloromethane, PL spectra in cyclohexane without $O_2$, PL and phosphorescence spectra in neat films for RTP-D1 (**a**) and RTP-D2 (**b**).

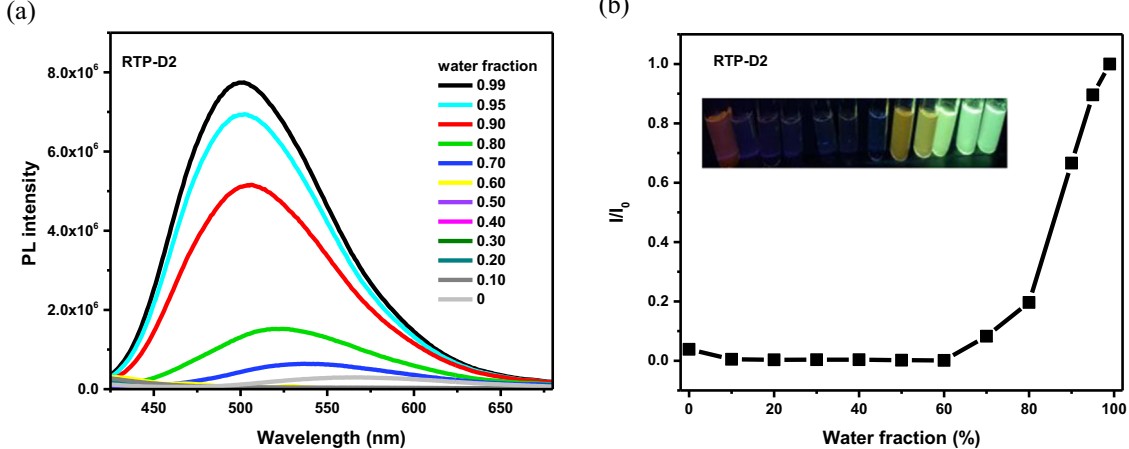

**Fig. 3 | AIE behavior of RTP-D2 in water/THF mixed solvents. a** Dependence of the PL spectra excited at 350 nm on the water fraction; **b** Relative emission intensity as a function of water fraction. Insets: PL images under UV light with the increasing water fraction from left to right.

about 7 nm relative to RTP-D1. The corresponding FWHMs of RTP-D1 and RTP-D2 are 87 nm and 91 nm, respectively. With regard to their PL spectra in dilute cyclohexane (Supplementary Fig. 11), the triplet excitons related RTP is believed to be involved in the emission to some degree, responsible for the large FWHMs. Indeed, a bright organic phosphorescence can be detected at room temperature for RTP-D1 and RTP-D2.

On one hand, they are found to give an interesting aggregation-induced emission (AIE). Taking RTP-D2 as an example (Fig. 3), the PL has a significant dependence on the water fraction ($f_w$) in THF. At the beginning, RTP-D2 seems to be almost non-emissive in pure THF

solution. After the addition of water, the relative intensity is greatly enhanced with the increasing $f_w$ from 70% to 99%. As mentioned above, RTP contributes to the PL of RTP-D2. Ongoing from solution to aggregate, the negative non-radiative decay from triplet excitons could be effectively restricted, thus generating an AIE behavior for RTP-D2.

On the other hand, time-resolved emission spectra (TRES) at room temperature and temperature-dependent transient PL spectra were recorded to demonstrate the RTP nature of RTP-D2. At a short delay time of 0.103 ns, the RTP-D2 neat film reveals a distinct dual PL including a major emission peaked at 471 nm and a shoulder appeared

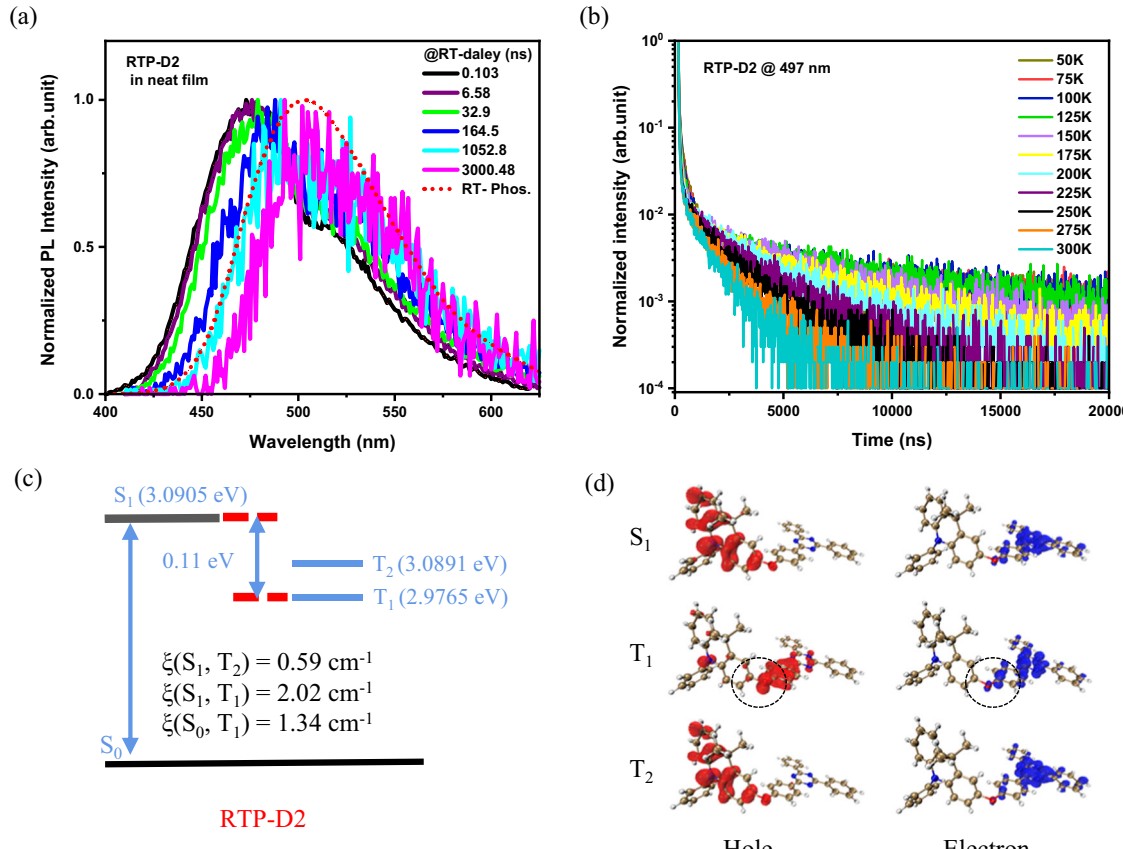

**Fig. 4 | RTP nature of RTP-D2 in neat film and theoretical calculations. a** Time-resolved emission spectra compared with the corresponding phosphorescent spectrum at room temperature; **b** Temperature-dependent transient PL spectra detected at 497 nm; **c** Energy level alignment together with the spin-orbit coupling matrix elements; **d** Hole and electron distributions of $S_1$ and $T_n$ below $S_1$.

at 500–550 nm (Fig. 4a). When the delay time is further up to 3000.48 ns, the shoulder becomes dominant in the whole PL, which matches well with the RTP spectrum. Noticeably, the TRES collected at 3000.48 ns can be well fitted with just one GaussMod, whose corresponding intensity is found to be increased distinctly from room temperature to 100 K (Supplementary Figs. 12 and 13). In addition, RTP-D2 possesses an obvious delayed component, which turns out to be gradually decreased as the temperature grows (Fig. 4b). The same trend is also observed for the steady-state PL spectra of RTP-D2 (Supplementary Fig. 14). All these observations clearly illustrate that RTP-D2 does achieve organic phosphorescence at room temperature.

To exclude thermally activated delayed fluorescence (TADF), the detected wavelength used for the transient PL measurement is intentionally set to be 416 nm in order to remove the RTP contribution (Supplementary Fig. 15). If there is TADF, you know, the delayed fluorescence would occur and increase with the increasing temperature[35]. In fact, only prompt fluorescence without any delayed one is observed, which is nearly insensitive to the temperature (Supplementary Fig. 15b). Therefore, we can deduce that the PL of RTP-D2 consists of prompt fluorescence and RTP, where not TADF but RTP is responsible for the delayed component. Following a Bigaussian fitting, the populations of prompt fluorescence and RTP are taken to be 29.4% and 70.6%, respectively, in the whole PL of RTP-D2 (Supplementary Fig. 16).

Similar behaviors are observed for RTP-D1 (Supplementary Figs. 17–21). In all, both RTP-D1 and RTP-D2 show an aggregation-induced organic RTP due to the characteristic D-O-A geometry. Nevertheless, the substituents on acridine seem to have some effect on their photophysical properties. With respect to RTP-D1 containing a methyl substituent ($\Phi_{PL} = 54.9\%$, $\tau_P = 488.8$ ns), RTP-D2 containing a phenyl substituent displays a higher PL quantum yield (PLQY) of 77.5% and a longer RTP lifetime of 641.4 ns (Supplementary Fig. 22).

**Theoretical simulation**

To further understand the relationship between the D-O-A geometry and the photophysical properties, theoretical simulation of RTP-D1 and RTP-D2 was then performed using Gaussian 09 package at a B3LYP/6-31G(d) level. With RTP-D2 as an example, the spin-orbital coupling matrix element (SOCME) between $S_1$ and $T_1$ is calculated to be 2.02 cm$^{-1}$, nearly quadrupled the value between $S_1$ and $T_2$ (Fig. 4c). The difference is understandable when considering the hole-electron distribution of RTP-D2. As one can see in Fig. 4d, $S_1$ and $T_2$ have a similar hole distribution on acridine and electron distribution on triazine. However, as for $T_1$, the hole is mainly localized on triazine and partly extended to the oxygen bridge. In this case, the n-orbital of the oxygen bridge is found to be responsible for the hole-electron distributions of $S_1$ and $T_1$ via a p–π conjugation with either D or A. According to the El-Sayed rule[36], such a hybridization of n and π orbitals could favor the ISC process. Most importantly, the hole migration from $S_1$ to $T_1$ involves the movement from D to A. From this point of view, the large dihedral angle between D and A (78.11° for RTP-D2 in Supplementary Fig. 23) may allow enough orbital angular momentum change to compensate the spin angular momentum[27]. Therefore, the spin flipping from $S_1$ to $T_1$ becomes more allowed than that from $S_1$ to $T_2$. Subsequently, the radiative decay from $T_1$ to $S_0$ is also facilitated because the n-orbital from the oxygen atom does contribute to the hole-electron distribution of $T_1$. And this agrees well with the large SOCME between $T_1$ and $S_0$ (1.34 cm$^{-1}$).

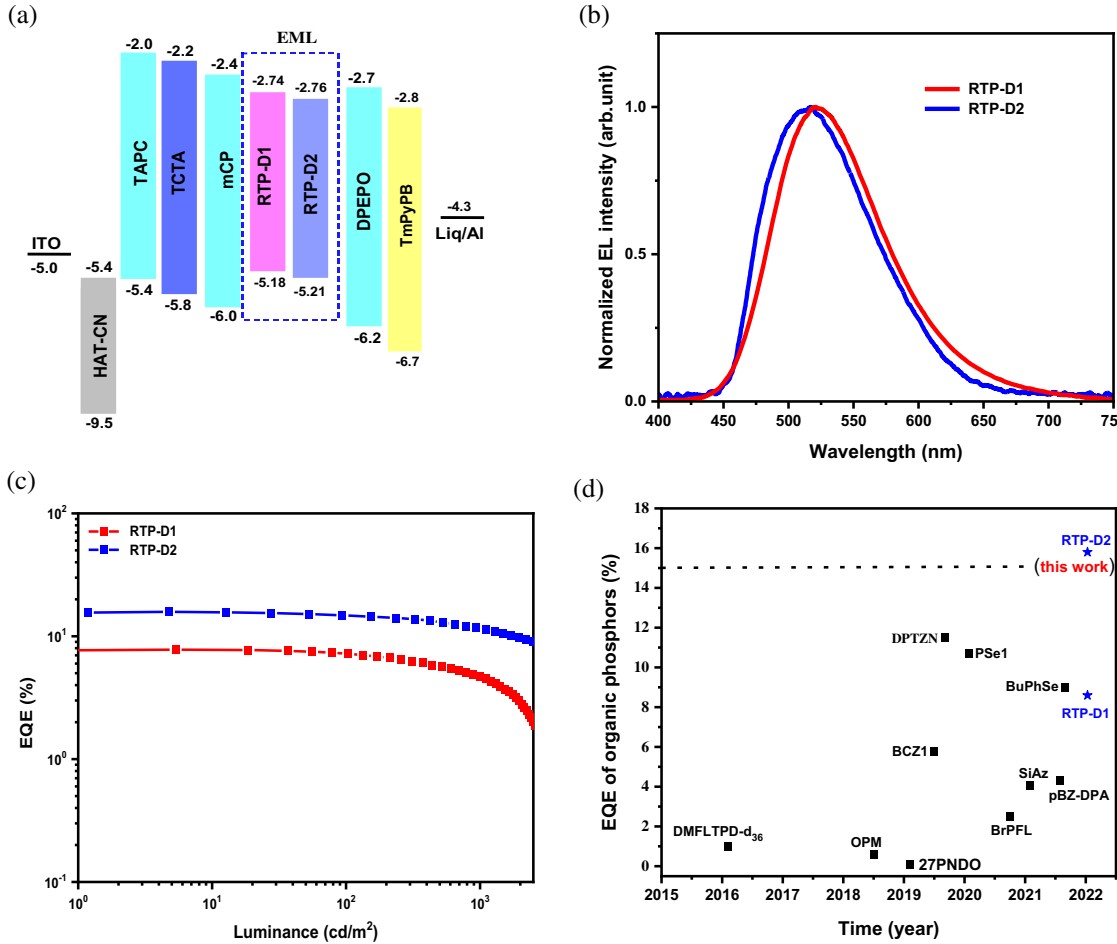

**Fig. 5 | Non-doped device performance for RTP-D1 and RTP-D2. a** Device structure and the related energy level diagram; **b** EL spectra at 6 V; **c** EQE as a function of luminance; **d** Efficiency comparison between this work and previously reported organic RTP emitters.

On the other hand, singlet excitons could not be converted to triplet excitons completely due to the moderate ISC rate constant ($k_{ISC}$) of $2.2–4.3 \times 10^7\,s^{-1}$. (Supplementary Table 2). There leaves a singlet exciton residue and thus a weak fluorescence (Supplementary Figs. 24 and 25). Meanwhile, the molecules are able to rotate around the C–O single bond (both D and A units), changing its conformation. In solution, the corresponding non-radiative decay plays an important role on the quenching of most triplet excitons. Hence only a weak RTP is detected besides fluorescence, resulting in a dual PL profile in the absence of $O_2$. Ongoing from solution to aggregates (or neat films), a rigid environment is provided to suppress such a rotation, so that the unwanted non-radiative decay of $T_1$ could be effectively eliminated. As a consequence, RTP is enhanced significantly, and dominates the whole PL. That is, RTP-D2 behaves an interesting aggregation-induced phosphorescence at room-temperature benefitting from the characteristic D-O-A geometry, in which the oxygen linkage is capable of strengthening the spin–orbital coupling to favor the ISC process and subsequent phosphorescence channel[32].

The same situation is also suitable for RTP-D1 (Supplementary Fig. 26). However, it shows a lower $S_1$-to-$T_1$ SOCME of 0.39 cm$^{-1}$ than that of RTP-D2 (2.02 cm$^{-1}$). The trend correlates well with the reduced $k_{ISC}$ ($2.2 \times 10^7\,s^{-1}$ for RTP-D1 Vs $4.3 \times 10^7\,s^{-1}$ for RTP-D1) and thus poorer phosphorescence for RTP-D1 ($\Phi_{PL}$ of 54.9%) relative to RTP-D2 ($\Phi_{PL}$ of 77.5%). Albeit this, they both obtain fast microsecond phosphorescent lifetimes[24,26], which may be tentatively attributed to the large dihedral angle between D and A in the D-O-A-based organic phosphors. Further experiments should be performed to verify this hypothesis, but they are now beyond the aim of this work.

## Aggregation-induced electrophosphorescence based on non-doped OLEDs

In terms of their good electroactivity, AIE and RTP features, RTP-D1 and RTP-D2 may have an efficient aggregation-induced electrophosphorescence in the challenging EL application. To demonstrate this point, non-doped OLEDs were fabricated with a device configuration of ITO/HATCN (3 nm)/TAPC (35 nm)/TCTA (5 nm)/mCP (5 nm)/EML (15 nm)/DPEPO (5 nm)/TmPyPB (45 nm)/Liq (1 nm)/Al (150 nm) (Fig. 5a and Supplementary Fig. 27). Here, 1,4,5,8,9,11-hexaaza-triphenylenehexacarbonitrile (HATCN), 4,4′-cyclohexylidenebis[N,N-bis(4-methylphenyl)aniline] (TAPC), 4,4′,4″-tris(carbazol-9-yl)-triphenylamine (TCTA) and 1,3-bis(carbazol-9-yl)benzene (mCP), bis[2-(diphenylphosphino) phenyl] ether oxide (DPEPO), 1,3,5-tri(m-pyrid-3-ylphenyl) benzene (TmPyPB) and 8-Hydroxyquinolinolato-lithium (Liq) are selected as the hole injection layer, hole-transporting layer, exciton blocking layer, electron transporting layer and electron injection layer, respectively. And the emitting layer (EML) is only composed of RTP-D1 or RTP-D2 by itself.

The EL spectra, EQE as a function of luminance and comparison with previous organic phosphors are plotted in Fig. 5, and the data are listed in Table 2. Similar to their PL counterparts, the EL spectra of RTP-D1 and RTP-D2 seem to be very broad with large FWHMs of 97 and 99 nm (Fig. 5b), respectively. With RTP-D2 as an example, its transient EL spectrum displays an obvious delay after switching off the electrical pulse (Supplementary Fig. 28), which is well consistent with the transient PL. Moreover, the EL profile keeps almost unchanged when the temperature varies. If an obvious TADF is involved, there would be a hypsochromic shift with the increasing temperature[37]. Therefore,

**Table 2 | Summary of the non-doped device performance of RTP-D1 and RTP-D2**

| EML | $V_{on}^a$ [V] | $CE^b$ [cd/A] | $PE^b$ [lm/W] | $EQE^b$ [%] | FWHM [nm] | $CIE^c$ [x, y] |
|---|---|---|---|---|---|---|
| RTP-D1 | 3.2 | 24.7/14.7 | 24.4/6.6 | 8.6/4.7 | 97 | (0.17, 0.59) |
| RTP-D2 | 2.8 | 45.8/35.5 | 50.4/21.4 | 15.8/11.7 | 99 | (0.16, 0.55) |
| RTP-D2: S-Cz-BN (2 wt%) | 2.7 | 54.0/37.2 | 63.0/25.4 | 26.4/18.1 | 26 | (0.12, 0.45) |

CE current efficiency, PE power efficiency, EQE external quantum efficiency.
[a]Turn-on voltage at 1 cd/m².
[b]Data at maximum and 1000 cd/m².
[c]Data at a driving voltage of 6 V.

TADF can be reasonably excluded, and the EL emission originates mainly from the RTP contribution. According to the literature[30], the RTP populations in the EL process are estimated to be about 92-93% for RTP-D1 and RTP-D2 (Supplementary Figs. 29 and 30). Owing to the different pathways of triplet excitons generation, reasonably, the values are much higher than those in the PL process (66.9% for RTP-D1 and 70.6% for RTP-D2). Albeit this, an intensive greenish-blue electrophosphorescence is successfully achieved for both RTP-D1 and RTP-D2, giving CIE coordinates of (0.17, 0.59) and (0.16, 0.55), respectively. Meanwhile, the maximum EQEs of RTP-D1 and RTP-D2 are obtained to be 8.6% and 15.8%, respectively (Fig. 5c, Supplementary Figs. 31 and 32). Noticeably, a small efficiency roll-off is observed at high luminance owing to their short phosphorescent lifetimes at a microsecond level. For example, the EQE of RTP-D2 is found to gently decay to 11.7% at a high luminance of 1000 cd/m². To our knowledge, the performance is the highest ever reported for organic RTP emitters (Fig. 5d and Supplementary Table 3)[22–31].

As discussed above, the conformation caused by the rotation around the C–O single bond has a significant influence on the photophysical properties. Here doping is adopted to control the conformation during the OLEDs fabrication. To evaluate this, the device performance dependence on the doping concentration is also investigated for RTP-D2 in mCP (Supplementary Figs. 33–37 and Supplementary Table 4). Because of the aforementioned HOMO and LUMO separation, RTP-D2 has a more than doubled dipole moment compared with mCP (Supplementary Fig. 38). Ongoing from doped to non-doped films, the rotation is anticipated to be prohibited due to the enhanced electrostatic interactions between RTP-D2. As a result of the reduced $T_1$ nonradiative decay, a distinctive aggregation-induced electrophosphorescence is achieved for RTP-D2, whose maximum EQE is monotonically increased with the increasing doping concentration. The trend is quite different from the previously reported D-O-A polymer, which gives a peak EQE of 9.7% at a doping concentration of 15% in the doped device (Supplementary Fig. 39). At present, we could tentatively deduce that the rotation around the C–O single bond is much easier in small molecule than in polymer, thus leading to the aggregation-induced electrophosphorescence for RTP-D2. It should be noted that the optimized device efficiency of RTP-D2 is achieved based on a non-doped configuration, where the difficult selection of the appropriate host and the tedious control of the dopant concentration could be avoided to simplify the device fabrication. These experimental facts clearly highlight the superiority of the D-O-A geometry in the successful realization of efficient aggregation-induced organic room-temperature electrophosphorescence.

### Host-free sensitization for multiple resonance emitter

You know, multiple resonance (MR) emitters have revolutionized OLEDs due to their capability to realize narrowband emissions for the next-generation wide-color gamut displays[38]. Although most of them exhibit TADF, their reverse intersystem crossing (RISC) rates are not sufficiently high, resulting in poor device efficiency and significant efficiency roll-off[39]. To solve this problem, a sensitizer combined with a wide bandgap host is usually introduced to assist in the harvesting of triplet excitons[40]. Albeit the success, the adoption of a ternary EML means a more complicated and tedious doping process. For simplification, therefore, much effort should be paid to develop a host-free sensitization, where only two components including the sensitizer and emitter constitute a binary EML, and the wide bandgap host is not demanded any longer.

Considering the obtained strong aggregation-induced electrophosphorescence, RTP-D2 shows a great potential in the host-free sensitization for MR emitters. To this end, S-Cz-BN[39] is selected as the MR dopant because there is a good overlap between the absorption of S-Cz-BN and the PL of RTP-D2 (Supplementary Fig. 40). Meanwhile, the energy levels of RTP-D2 are determined to be 2.94 eV for $S_1$ and 2.73 eV for $T_1$ (Supplementary Fig. 41), higher than that of S-Cz-BN ($S_1$ of 2.54 eV and $T_1$ of 2.36 eV). These requirements ensure the efficient Förster energy transfer (FET) from RTP-D2 to S-Cz-BN. Following the same device configuration, hence S-Cz-BN is doped into RTP-D2 to evaluate the sensitization effect of RTP-D2 on S-Cz-BN. The doping concentration of S-Cz-BN is set at a low level of 2 wt.%, so that the unwanted Dexter energy transfer (DET) could be avoided.

As depicted in Fig. 6a, the EL profile turns out to be much sharper after doping. Moreover, the corresponding FWHM is down from 99 nm of RTP-D2 to 26 nm of RTP-D2: 2 wt.% S-Cz-BN. The observation suggests that S-Cz-BN is responsible for the narrowband EL due to the efficient FET from RTP-D2 to S-Cz-BN. Consequently, a state-of-art EQE of 26.4% is achieved together with a maximum current efficiency of 54.0 cd/A and a maximum power efficiency of 63.0 lm/W (Fig. 6b and Supplementary Fig. 42). Under electric excitation, 25% singlet excitons and 75% triplet excitons are first generated on RTP-D2. Then they are able to be converted to the $S_1$ state of S-Cz-BN via an effective FET process (Fig. 6c). In such a case, both singlet and triplet excitons could be fully harvested by S-Cz-BN, leading to the promising device efficiency. The result indicates the great potential of D-O-A-based organic phosphors, which are capable of host-free sensitization for MR emitters.

## Discussion

According to a proposed D-O-A molecular design, a class of organic RTP emitters named RTP-D1 and RTP-D2 has been developed based on acridine as the electron donor, triazine as the electron acceptor and oxygen as the bridge between them. Since the involved oxygen bridge between D and A units can strengthen the spin–orbital coupling to promote the ISC process and subsequent phosphorescence channel, both RTP-D1 and RTP-D2 in neat films display a bright aggregation-induced organic room-temperature electrophosphorescence by themselves. Simultaneously, they can be used as host-free sensitizers for MR emitter, revealing an efficient narrowband emission (EQE = 26.4% and FWHM = 26 nm). We believe this work provides us a promising strategy for the design of efficient metal-free organic phosphors, and will pave the way to their potential applications in OLEDs.

## Methods

### Material characterization

¹H and ¹³C nuclear magnetic resonance (NMR) spectra were performed on a Bruker Avance NMR spectrometer. Electrospray Ionization Mass Spectroscopy (ESI-MS) was measured on WATERS CORPORATION instrument. Elemental analyses were recorded by a Bio-Rad elemental analysis system. Thermal gravimetric analysis (TGA) and differential scanning calorimetry (DSC) were recorded on TA-TGA55 and TA-DSC25 under nitrogen atmosphere at a heating rate of 10 °C/min, respectively. Cyclic voltammetry (CV) was measured on a CHI660a electrochemical analyzer using ferrocene/ferrocenium (Fc/Fc⁺) as the

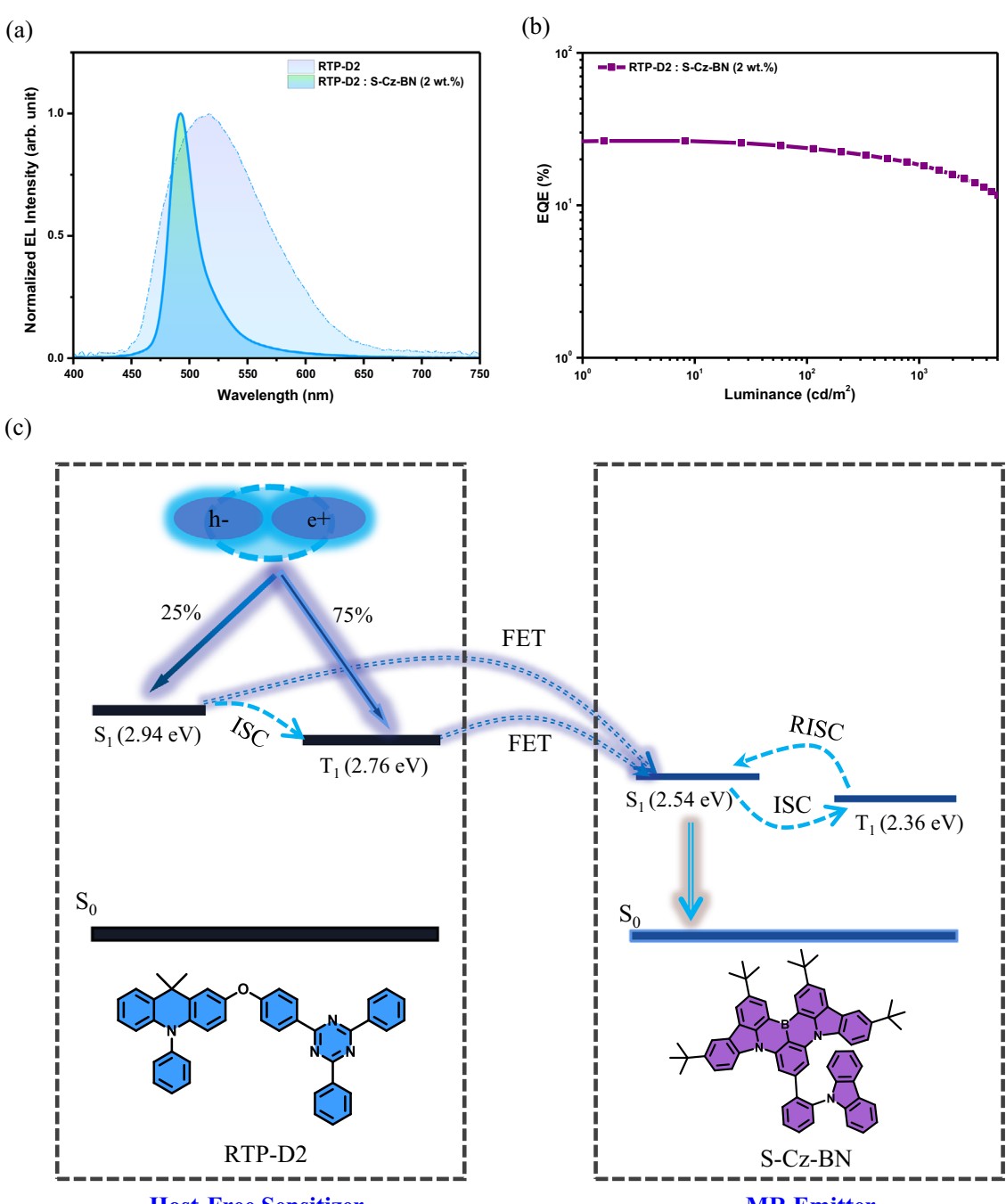

**Fig. 6 | Host-free sensitization of MR dopant S-Cz-BN using RTP-D2. a** Comparison of the EL spectra between RTP-D2 and RTP-D2:S-Cz-BN (2 wt%); **b** EQE as a function of luminance; **c** Mechanism of host-free sensitization.

reference and n-Bu$_4$NClO$_4$ (0.1 M) as the supporting electrolyte. The HOMO and LUMO energy levels were calculated by the equation: HOMO (or LUMO) = −e [$E_{onset, ox}$ (or $E_{onset, red}$) + 4.8 V], where $E_{onset, ox}$ is the onset value of the first oxidation wave and $E_{onset, red}$ is the onset value of the first reduction wave. UV–Vis absorption spectra were measured with a Perkin-Elmer Lambda 35 UV–Vis spectrometer. The steady-state PL spectra, room-temperature phosphorescent spectra and $\Phi_{PL}$s were measured on a HORIBA FL3C-111 spectrofluorometer equipped with an integrating sphere and a liquid nitrogen-cooled optical cryostat (Optistat DNV, Oxford Instruments) with an ITC503S temperature controller. Time-resolved emission spectra (TRES) were measured on a HORIBA Delta Flex modular fluorescence lifetime system equipped with a NanoLED-375 source. The transient PL spectra

were recorded in vacuum using Edinburgh fluorescence spectrometer (FLSP-980).

**Theoretical simulations**
Density functional theory (DFT) and time-dependent DFT (TD-DFT) calculations were performed using the Gaussian 09 program packages[41] to calculate the frontier molecular orbital distributions, and energies of the key transitions. First, the geometries in the ground state were directly obtained from the single crystals of RTP-D1 and RTP-D2. Second, the excited states (S$_1$ and T$_n$ below S$_1$) energies, spin-orbit coupling matrix elements of S$_1$-to-T$_n$ and T$_1$-to-S$_0$ were calculated using TD-DFT at a B3LYP/6-31G(d) level[42,43] according to the crystal structures. Third, hole and

electron analysis[44] were performed using Multiwfn program[45] and visualized by VMD software[46].

## Single-crystal X-ray diffraction

The single-crystal X-ray diffraction experiments were carried out using a Bruker D8 VENTURE area detector diffractometer with graphite monochromator Mo Khromatorchro ($\lambda = 0.71073$ Å). All non-hydrogen atoms were assigned with anisotropic displacement parameters, whereas hydrogen atoms were placed at calculated positions theoretically and included in the final cycles of refinement in a riding model along with the attached carbons. The packing modes and dihedral angles were exhibited using Mercury 2022.2.0 free from Cambridge Crystallographic Data Center. The ORTEP drawings were displayed in ORTEP-3[47]. Crystallographic data for the structural analyses have been deposited with Cambridge Crystallographic Data Center (CCDC). The CCDC reference is 2175448 for RTP-D1 and 2175449 for RTP-D2.

## Device fabrication and measurements

Before devices fabrication, the ITO substrates with a sheet resistance of 15 Ω per square were cleaned by sequential ultra-sonication in detergent, deionized water, acetone, ethanol, and then exposed to UV-Ozone for 15 min. After being transferred into a vacuum chamber, all material layers were deposited by vacuum evaporation in a vacuum chamber with a base pressure of $<3 \times 10^{-5}$ Pa. As for host-free sensitization, S-Cz-BN is doped into RTP-D2 to constitute a binary EML at a low content of 2 wt.%. The current density-voltage characteristics were performed using an HP4140B picoammeter. And the luminance and electroluminescence (EL) spectra were recorded by Minolta LS-110 Luminance meter and Ocean Optics USB-4000 spectrometer, respectively. EQE was calculated from the EL spectrum, luminance and current density assuming a Lambertian emission distribution. All the measurements were carried out at room-temperature under ambient conditions without device encapsulation. The temperature-dependent EL spectra were measured using a liquid nitrogen-cooled optical cryostat (Optistat DNV, Oxford Instruments) with an ITC503S temperature controller. The transient EL spectra were measured using KEYSIGHT DSO1012A oscilloscope, equipped with regulated DC power supply of LINI-UTP3313TFL-11 and VICTOR DDS signal generator counter.

## Data availability

The data that support the findings of this study are presented in the Supplementary Information. The source data underlying Figs. 1b and 2–6 are provided in the Source Data files with this paper or available from the corresponding author on request. Crystallographic data for structures reported in this paper are available free of charge from the Cambridge Crystallographic Data Centre under deposition number CCDC 2175448 (RTP-D1) and 2175449 (RTP-D2) via www.ccdc.cam.ac.uk. Source data are provided with this paper.

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

## Acknowledgements
The authors acknowledge the financial support from the National Natural Science Foundation of China (Nos. 52273198 and 51873205, J.D.), and the Scientific Research Foundation for Introduced Talents of Yunnan University (CZ21623201, J.D.). L. Xu acknowledges the Research Innovation Fund for Graduate Students of School of Chemical Science and Technology, Yunnan University. We also thank Advanced Analysis and Measurement Center of Yunnan University for the assistance with instrumentation.

## Author contributions
L.X. performed the synthesis, characterization, and analysis of the RTP emitters. N.S. participated the characterization of the TRES. Y.M. and C.S. performed the device fabrication. N.S. participated the analysis of device performance. Y.Z. and L.D. offered the MR dopant S-Cz-BN and discussed the concept of host-free sensitization. Z.-H.L. helped the characterization and discussion of the transient EL. J.D. supervised and directed the project. L.X. and J.D. wrote and revised the paper.

## Competing interests
L.X. and J.D. are inventors on a provisional patent application related to the described work. The other authors declare no competing interests.
