## [Peer Review File · Nature Communications]

D-O-A based organic phosphors for both aggregation-induced electrophosphorescence and host-free sensitizationReviewers' Comments:

Reviewer #1:

Remarks to the Author:

In this manuscript, Shi, Zhang, Ding, and their coauthors disclose the development of purely organic room-temperature phosphorescence (RTP) materials based on D-O-A skeleton. The synthesized materials (RTP-D1 and RTP-D2) displayed CT emission band at around 500 nm in solutions and neat films. The compounds showed aggregation-induced enhanced emission (AIEE) behavior, which is beneficial for hot-free OLED devices. Photophysical analysis of the compound in neat films indicated RTP emission from the T1 state, which was supported by theoretical calculations. The host-free OLED devices fabricated with the synthesized emitting materials showed a high EQE up to 11.7%. Furthermore, the authors executed sensitization for host-free MR emitters to achieve narrow-emission band OLEDs with an excellent EQE. Overall, the manuscript is well organized and the discussion is partly sound. However, I am skeptical about the novelty according to the reasons included in the questions and comments below, and therefore, I cannot recommend the manuscript for publication in Nature Communications. Instead, I would recommend that the authors submit to more specific journals such as Journal of Materials Chemistry C and Chemistry of Materials, after the authors revise the manuscript accordingly.

#1. First of all, the authors do not explain the novelty of material design of RTP-D1 and D2. The authors have already reported a paper in which RTP polymers having a very similar structure with RTP-D1 as a monomer unit in 2021 (ref. 31). If they appeal the novelty of this material, the authors must explain why this monomeric structured molecule is superior to the previously reported polymeric materials based on experiments and theoretical studies. Even so, significant advancement in terms of molecular design is not recognized in the current manuscript. I would suggest that the authors collect evidence for proving the difference in performances and re-submit as a Full Paper to a more specific journal.

#2. The authors discussed the RTP characteristics in PL study, but no analysis was done for OLED devices. On one hand, the authors claim that the performance is the highest ever reported for organic RTP emitters. This phrase is overexaggerated. How do the authors know the EL emission is RTP not TADF? The authors must prove it, if they want to claim that.

#3. The authors calculate the RTP-D1 and D2 and discuss the mechanism of RTP (Figure 4). However, the molecules can rotate around the C-O single bond (both donor and acceptor), changing its conformation. The conformation must give significant influence on the excited state energy and molecular orbitals, which further affect RTP efficiency. When OLEDs are fabricated, how do the authors control the conformation?

Reviewer #2:

Remarks to the Author:

Metal free organic RTP is an attractive mechanism for electroluminescence generation in OLEDs. Unfortunately, the number of metal free organic RTP materials that are capable of efficient EL generation in OLEDs is relatively low, mainly due to the long lifetime associated with RTP, which is susceptible to cause EL quenching caused by interactions between charge carriers and long-lived excited states.

In this work, the authors report the design of two RTP emitters formed by an acridine donor unit linked to a triazine acceptor through an oxygen bridge. The impact of this work is slightly diminished because the emission of these new compounds is in the already crowded green spectral region for OLED emitters, and because using an oxygen bridge in D-O-A emitters to promote faster ISC is not entirely new. Some of these authors have recently published the application of a metal-free RTP

material in OLEDs (ref 31) with a D-O-A structure. However, this work is still interesting and may be worth publishing in Nat. Com., since the two new compounds RTP-D1 and RTP-D2 demonstrated very good OLED performance with EQE above 8% and 15%, respectively, with minimal roll-off up to 1000 Cd/m² in host free OLEDs.

I have some concerns related with the origin of the delayed emission that the authors attribute exclusively to RTP. Looking at Fig 4(a), the TRES collected with 1052.8 ns delay time seems to show two separate bands, one peaking around 480 nm and the other peaking at 530 nm. Both these emission bands are contained within the "RT-Phosp" band. However, if the emission is totally due to RTP, and the RTP spectrum shows no vibrational progression, why are two separate bands observed in the time-resolved spectra? To clarify this issue, a deconvolution of the TRES-RTP spectra collected around 1000 ns delay time should be performed as a function of temperature, using two Gaussian curves (as it was done for the RT-PL steady-state data in Fig S11). If the RTP-TRES can be fitted with just one Gaussian, then we can say with reasonable assurance that the delayed emission is purely due to RTP. However if two Gaussians are needed to fit the RTP-TRES, then TADF might be involved in the delayed emission, and this will be confirmed by the temperature dependence of these two bands. Equal procedure must be performed for the TRES-RTP data shown in Fig S9, to check the behaviour at longer delay times where TADF might be less prevalent. I also would like to see how the delayed emission is affected by the presence of oxygen, so the triplet state origin of the delayed emission can be confirmed.

I therefore recommend major revision before the manuscript can be accepted for publication.

Point-by-point response to the comments

Response to Reviewer #1

Q1: First of all, the authors do not explain the novelty of material design of RTP-D1 and D2. The authors have already reported a paper in which RTP polymers having a very similar structure with RTP-D1 as a monomer unit in 2021 (ref. 31). If they appeal the novelty of this material, the authors must explain why this monomeric structured molecule is superior to the previously reported polymeric materials based on experiments and theoretical studies. Even so, significant advancement in terms of molecular design is not recognized in current manuscript. I would suggest that the authors collect evidence for proving the difference in performances and re-submit as a Full Paper to a more specific journal.

A: (1) Thanks a lot for your good advice. After reevaluation, we believe that the novelty of this manuscript lies in three aspects as followed:

① **Based on the D-O-A structure, room-temperature phosphorescence (RTP) is successfully realized at a small molecular level:** As you point out, we previously demonstrated a D-O-A type polymer showing efficient RTP (*Angew. Chem. Int. Ed.* **2021**, *60*, 2455). However, X. –H. Zhang et al. reported a D-O-A type small molecule, which showed a thermally activated delayed fluorescence (TADF) via intermolecular charge transfer (*Angew. Chem. Int. Ed.* **2018**, *57*, 9480). The two results seem to be contradictory about the D-O-A structure. Therefore, it still remains a doubt whether the D-O-A structure could realize efficient RTP at a small molecular level, when given the difference between polymers and small molecules including conjugation

length, intermolecular interactions, rotating possibility of C-O single bond and etc.

In this work, two metal-free organic phosphors named RTP-D1 and RTP-D2 have been newly developed by using acridine as D, triazine as A and O as the bridge between D and A. Both of them show an exciting aggregation-induced organic room-temperature electrophosphorescence by themselves, revealing a record-high EQE of 15.8% (45.8 cd/A, 50.4 lm/W) for non-doped OLEDs. The results clearly remove the above-mentioned doubt.

Compared with the difference between this work and X. -H. Zhang's report, we think that the O position between D and A may have a significant influence on their luminescent mechanism. So we design and synthesize three compounds by varying the O position. Thanks for the obtainment of their single crystals, further work is under way and will be published elsewhere.

② From aggregation-caused quenching in macromolecule to aggregation-induced electrophosphorescence in small molecule: According to our previous report, the D-O-A polymer shows a maximum EQE of 9.7% at a 15% doping concentration (doped device), indicative of the aggregation-caused quenching in OLEDs. However, the newly-developed D-O-A small molecule exhibits a different aggregation-induced electrophosphorescence, revealing a maximum EQE above 15% in nondoped device. This is quite interesting, because the difficult selection of the appropriate host and the tedious control of the dopant concentration could be avoided to simplify the device fabrication.

Figure S36. (a) Comparison between D-O-A based polymer and small molecule; (b) Maximum EQE as a function of doping concentration for RTP-D2 in mCP.

③ From host + TADF sensitizer + MR emitter to aggregation-induced

electrophosphorescence + MR emitter (host-free sensitization): Multiple resonance (MR) emitters have revolutionized OLEDs due to their capability to realize extremely narrowband emissions for the next-generation wide-color gamut displays. In general, a TADF sensitizer combined with a wide bandgap host is usually introduced to assist in the harvesting of triplet excitons for the MR dopant. Albeit the success, the adoption of a ternary emitting layer means the more complicated and tedious doping process.

To solve this problem, a host-free sensitization is demonstrated by taking advantage of the characteristic aggregation-induced electrophosphorescence of RTP-D2. And an efficient narrowband emission is achieved with a state-of-art EQE of 26.4% and a small full width at half maximum (FWHM) of 26 nm. The results highlight its great potential on the host-free sensitization for MR emitters.

Figure 6. Host-free sensitization of MR dopant S-Cz-BN using RTP-D2: (a) EL spectra without and with S-Cz-BN; (b) EQE as a function of luminance; (c) Mechanism of host-free sensitization.

(2) The discussion and data about the above three aspects are involved in the revised manuscript. As pointed out by reviewer #2, this work is still interesting and worth publishing in Nat. Commun. (**Marked as yellow**)

Q2: The authors discussed the RTP characteristics in PL study, but no analysis was done for OLED devices. On one hand, the authors claim that the performance is the highest ever reported for organic RTP emitters. This phrase is overexaggerated. How do the authors know the EL emission is RTP not TADF? The authors must prove it, if they want to claim that.

A: (1) Thanks for your doubt about the EL nature. You know, PL and EL only differ in the exciton generation route. As for PL, singlet excitons are firstly generated by photo excitation, followed by an ISC process to produce triplet excitons. As for EL, holes and electrons are injected and transported to recombine together in order to form singlet and triplet excitons directly. Owing to the temperature influence on the charge transporting/trapping and thus EL intensity, there lack effective analysis methods for OLEDs to directly verify the EL nature at present. According to previous literatures about fluorescence, metal-containing phosphorescence, TADF and RTP, the EL nature is believed to be the same as the PL if they have similar emission spectra.

(2) In this work, the detailed PL study has demonstrated the RTP characteristics for D-O-A organic phosphors (RTP-D1 and RTP-D2). Most importantly, we note that the EL spectra are similar to the PL counterparts (Figure 2 and 5). From this point of view, we can deduce that the EL of RTP-D1 and RTP-D2 is mainly from RTP.

(3) Moreover, we try to find a new way to directly verify the EL nature in OLEDs. According to our previous work, when only RTP is involved, the PL intensity is gradually decreased with the increasing temperature, and the emission maxima remain nearly unchanged or are red-shifted due to the rigidochromic effect. Otherwise, if an obvious TADF is involved, the emission maxima are blue-shifted significantly with the increasing temperature, accompanied by the intensity decrease. Unlike RTP from the low-energy T_1 , TADF is from the high-energy S_1 and becomes more and more considerable as temperature grows, thus leading to a distinct hypsochromic shift.

On the basis of this experimental fact, we measure the EL spectra of RTP-D2 at different temperature (Although the intensity variation is meaningless, the spectral shift is worthy to be studied). In fact, no hypsochromic shift is observed with the increasing temperature. Therefore, TADF can be reasonably excluded, and the EL

emission originates mainly from the RTP contribution. The related discussion is added into the revised manuscript as well as Figure S25. (Marked as grey)

Q3: The authors calculate the RTP-D1 and D2 and discuss the mechanism of RTP (Figure 4). However, the molecules can rotate around the C-O single bond (both donor and acceptor), changing its conformation. The conformation must give significant influence on the excited state energy and molecular orbitals, which further affect RTP efficiency. When OLEDs are fabricated, how do the authors control the conformation?

A: (1) Yes, I agree with your opinion. The conformation caused by the rotation around the C-O single bond has a significant influence on the photophysical properties, leading to an interesting aggregation-induced phosphorescence in the PL process.

(2) In our current work (to be published), a newly-synthesized D-O-A organic phosphor RTP-D3 shows a dual emission, where RTP is easily distinguished from fluorescence. When doped into mCP, the ratio of RTP to fluorescence is gradually increased with the increasing doping concentration in the EL. Because of the HOMO and LUMO separation, RTP-D3 has a larger dipole moment than that of mCP. Ongoing from doped to nondoped films, the rotation is anticipated to be prohibited due to the enhanced electrostatic interactions between RTP-D3. Consequently, the unwanted non-radiative decay of T_1 is reduced, leading to the improved RTP contribution in the EL. So when OLEDs are fabricated, doping can be used to control the conformation.

(3) Based on such a finding, herein RTP-D2 is doped into mCP, and the corresponding device performance dependence on the doping concentration is investigated in detail. It is found that the maximum EQE is monotonically increased with the increasing doping concentration. This indicates the aggregation-induced electrophosphorescence in the EL process, which may be tentatively ascribed to the variation of the rotation possibility. And the related discussion is added into the revised manuscript as well as Figure S30-S35 and Table S4. **(Marked as green)**

Table S4. Device performance dependence on the doping concentration of RTP-D2.

Doping ratio	V_{on}^a	CE ^b	PE ^b	EQE ^b
10 %	3.4	22.0 / 11.6	19.2 / 5.1	9.3 / 4.9
20 %	3.4	28.8 / 17.9	25.1 / 8.8	11.6 / 7.0
30 %	3.4	32.2 / 21.3	29.8 / 11.2	12.8 / 8.4
60 %	3	38.2 / 30.1	36.0 / 18.8	13.7 / 9.4
90 %	3	42.0 / 32.2	42.0 / 19.4	14.4 / 11.0
100 %	2.8	45.8 / 35.5	50.4 / 21.4	15.8 / 11.7

Response to Reviewer #2

Q1: I have some concerns related with the origin of the delayed emission that the authors attribute exclusively to RTP. Looking at Fig 4(a), the TRES collected with 1052.8 ns delay time seems to show two separate bands, one peaking around 480 nm and the other peaking at 530 nm. Both these emission bands are contained within the “RT-Phosp” band. However, if the emission is totally due to RTP, and the RTP spectrum shows no vibrational progression, why are two separate bands observed in the time-resolved spectra? To clarify this issue, a deconvolution of the TRES-RTP spectra collected around 1000 ns delay time should be performed as a function of temperature, using two Gaussian curves (as it was done for the RT-PL steady-state data in Fig S11). If the RTP-TRES can be fitted with just one Gaussian, then we can say with reasonable assurance that the delayed emission is purely due to RTP. However if two Gaussians are needed to fit the RTP-TRES, then TADF might be involved in the delayed emission, and this will be confirmed by the temperature dependence of these two bands.

A: (1) Thanks a lot for your carefulness. According to your suggestion, we fail to deconvolute the TRES spectrum at a 1052.8 ns delay time based on a Bigaussian fitting. Therefore, we do recheck the original data, and collect a curve at a longer delay time of 3000.48 ns. In this case, it only shows a major band.

(2) Despite this, as suggested, we still measure the TRES at different temperature, and perform the comparison for the curve collected at 3000.48 ns. It is found that the corresponding intensity is increased distinctly from room temperature to 100 K, an indicator of RTP covering a 450-600 nm range (Figure S11).

Figure S11. TRES for the RTP-D2 film: (a) at 100 K; (b) at 200 K; (c) at room temperature; (d) Intensity comparison of the curves at a delay time of 3000.48 ns.

(3) As suggested, the TRES collected at 3000.48 ns can be well fitted with just one GaussMod (Figure S12). Combined with the transient PL detected at 416 nm (Figure

S14b) and the temperature-dependent EL (Figure S25, see Q2 from Reviewer #1), TADF is reasonably excluded, and the delayed emission is purely due to RTP. The related discussion is provided in the revised manuscript. (Marked as pink)

Figure S12. GaussMod fitting for the TRES collected at 3000.48 ns: (a) at 100 K; (b) at 200 K; (c) at room temperature.

Q2: Equal procedure must be performed for the TRES-RTP data shown in Fig S9, to check the behaviour at longer delay times where TADF might be less prevalent.

A: In Figure S12 (Figure S9 in the former version), it is not TRES but steady-state PL. As suggested, the PL is measured with varied delay times. It is found that the PL profile is nearly independent on the delay time. So a 0.1 ms delay is used for the

measurement of phosphorescence spectra. (Marked as red)

Figure S13. (a) PL spectra at different delay times for the RTP-D2 film; (b) Temperature-dependent PL spectra (without a delay) for the RTP-D2 film; (c) Temperature-dependent phosphorescence spectra (with a 0.1 ms delay) for the RTP-D2 film. As one can see, the PL profile is nearly independent on the delay time. So a 0.1 ms delay is used for the measurement of phosphorescence spectra.

Q3: I also would like to see how the delayed emission is affected by the presence of oxygen, so the triplet state origin of the delayed emission can be confirmed.

A: (1) As suggested, the O₂ influence is provided both in neat films and in

cyclohexane solution. (Marked as sky-blue)

(2) In neat films, the PL is sensitive to O₂ obviously, confirming the triplet state origin of the delayed emission.

Figure S9. The O₂ dependence of the PL spectra in neat films for RTP-D1 (a) and RTP-D2 (b). The PL spectra of RTP-D1 and RTP-D2 are found to be sensitive to O₂, indicative of the triplet exciton origin to some degree.

(3) In cyclohexane solution, both RTP-D1 and RTP-D2 show two distinct emission bands in the absence of O₂. Assumed that the emission corresponding to the triplet excitons could be completely quenched by O₂ in dilute solution, the long-wavelength band is reasonably from the triplet exciton related RTP.

Figure S10. The O₂ dependence of the PL spectra in cyclohexane (10⁻⁵ mol L⁻¹) for RTP-D1 (a) and RTP-D2 (b). As one can see, both RTP-D1 and RTP-D2 show two distinct emission bands in the absence of O₂. Assumed that the emission corresponding to the triplet excitons could be completely quenched by O₂ in dilute solution, the long-wavelength band is reasonably from the triplet exciton related RTP.

Reviewers' Comments:

Reviewer #1:

Remarks to the Author:

The content has been significantly revised in this version. Yet, the questions raised by the reviewers have not been appropriately responded. For example, the authors still claim the EL is derived from RTP, although they do not obtain experimental proof of pure RTP from EL. Therefore, I cannot recommend the manuscript for publication in Nature Communications. Instead, I would recommend that the authors submit to more specific journals such as Journal of Materials Chemistry C and Chemistry of Materials, after the authors revise the manuscript accordingly.

Reviewer #2:

Remarks to the Author:

I am satisfied with the responses given to reviewers' questions. I think the authors did everything possible to confirm the RTP origin of the delayed emission, and while I am still sceptic about TADF not being involved, I have no other way to suggest how to distinguish between TADF and RTP as these may be contributing in different time ranges. Since based on Fig S13 c) it is clear that RTP plays a significant role in the luminescence of these compounds and the these are nondoped devices with reasonable good EQE, I am happy to recommend the manuscript to be accepted for publication in Nat. Comm. No further revision is needed on my side.

Reviewer #3:

Remarks to the Author:

The reported phenomena are indeed quite interesting. However, this reviewer does not think that the authors actually revealed the molecular feature responsible for the phenomena. There are many unclearly and even discrepant findings/explanations.

First of all, yes, the simple connection of D and A units with the oxygen bridge will add the n character. However, because the oxygen does not have a pi-bonding different from carbonyl the conventional moiety to mix the n and pi character, there is no n-pi to pi-pi or pi-pi to n-pi type El-Sayed rule satisfying feature in the molecular design. As the computational results show in Figure 4(d) the large difference in hole distribution between S1 and T2 with a large dihedral angle may allow enough orbital angular momentum change to compensate the spin angular momentum. However, this is not the case at all for RTP-D1. Then how should explain the observed phenomena from RTP-D1? More strangely, why does the D-O-A structure produce quite unprecedentedly fast microsecond phosphorescence?

The phosphorescence lifetime listed in Table 1 are very fast at 0.4 - 0.6 micro second about the same level of the organometallic phosphors, even though the delayed emission measurement was carried out at 100 micro seconds???. Where does the super efficient spin orbit coupling for this fast emission come from? This fast phosphorescence should not be sensitive to collisional quenching. Then how should explain the AIE? The same suspicion is on the oxygen quenching. Unlike purely organic phosphors having very slow triplet emission, fast phosphorescence emitters are not that sensitive to molecular triplet oxygen. However, the solution data in Figure S10 show almost complete quenching in the presence of oxygen.

When it comes to the emission lifetime and the delay time, there is another example of confusion. Fig S18 shows the emissions at the 100 micro second delay time. At 300K, the emission at about 500nm is still observed. However, in Figure S17 the emission lifetime at 300 looks to be 5 - 10 microsecond which shouldn't be detected at 100 microsecond delay time.

Another question is for the difference in the emission profiles of the solution and the neat film (for example, Figure S9 and S10). While the solution shows fluorescence and phosphorescence-like emission, the neat films emit mostly the phosphorescence peak at around 500nm. What boost so efficiently the intersystem crossing from the single to triplet in the neat film that is missing in the solution?

There are more questions and suggestions.

Figure 2. should show the PL spectra of the solutions in the panel (a). The PL spectra of the neat films are already in the panel (b).

Fig S16. The emission lambda max in THF is at around 550 nm??? If the molecular motion causes the quenching of the emission there shouldn't be emission lambda max change.

Figure 3 (a) should show whole spectrum from the excitation wavelength. What is the excitation wavelength? Why does the emission color (emission lambda max) gradually change? Shouldn't only the emission intensity increase?

Hoped to see a better balance in reference because two seminal papers having high citation numbers and demonstrating the very early stage of rational organic phosphor design should have been cited (Nat. Mater. 2009, 8, 747 & Nat. Chem. 2011, 3, 205).

Point-by-point response to the comments

Response to Reviewer #1

Q1: The content has been significantly revised in this version. Yet, the questions raised by the reviewers have not been appropriately responded. For example, the authors still claim the EL is derived from RTP, although they do not obtain experimental proof of pure RTP from EL. Therefore, I cannot recommend the manuscript for publication in Nature Communications. Instead, I would recommend that the authors submit to more specific journals such as Journal of Materials Chemistry C and Chemistry of Materials, after the authors revise the manuscript accordingly.

A: (1) Thanks a lot for your affirmation on the first revision.

(2) As for the EL mechanism, there may be a misunderstanding about the difference between PL and EL. You know, PL and EL only differ in the routes of exciton generation (see Figure). In the PL, singlet excitons are firstly formed by photo excitation, followed by a radiative decay to give fluorescence (1st Gen: fluorescence). If there is an efficient intersystem crossing (ISC) process, then singlet excitons can spin-flip to triplet excitons, leading to the production of phosphorescence (2nd Gen: metal-containing phosphorescence). Furthermore, if the singlet-triplet energy difference (ΔE_{ST}) is small enough to promote reverse ISC (RISC) under environmental heat, triplet excitons can be up-converted to singlet excitons, leading to thermally activated delayed fluorescence (3rd Gen: TADF).

In the EL, the injected holes and electrons are recombined to directly generate

single and triplet excitons with a statistical ratio of 1:3. As for 1st Gen, fluorescence is obtained from singlet excitons, whereas triplet excitons are quenched via non-radiative decay (NR). As for 2nd Gen, phosphorescence is obtained from triplet excitons, which are produced not only through the hole and electron recombination, but also through the ISC of single excitons. As for 3rd Gen, besides prompt fluorescence from singlet excitons, TADF is also obtained through the RISC from triplet to singlet excitons.

Figure. Mechanism similarity between PL and EL.

In all, the EL mechanism is reasonably proposed on the basis of PL for previously-reported fluorescence, metal-containing phosphorescence and TADF

systems. That is, PL and EL behave a similar mechanism in spite of their different exciton generation routes.

(3) As for the D-O-A based organic phosphors in this work, the room-temperature phosphorescence (RTP) origin of the PL has been fully confirmed (as pointed out by reviewer #2). Therefore, it is reasonable for us to propose the corresponding EL mechanism, which is mainly from RTP.

Response to Reviewer #3

Q1: First of all, yes, the simple connection of D and A units with the oxygen bridge will add the n character. However, because the oxygen does not have a pi-bonding different from carbonyl the conventional moiety to mix the n and pi character, there is no n-pi to pi-pi or pi-pi to n-pi type El-Sayed rule satisfying feature in the molecular design. As the computational results show in Figure 4(d) the large difference in hole distribution between S1 and T2 with a large dihedral angle may allow enough orbital angular momentum change to compensate the spin angular momentum. However, this is not the case at all for RTP-D1. Then how should explain the observed phenomena from RTP-D1?

A: (1) In RTP-D2: Yes, you are right. Unlike the conventional carbonyl, the oxygen atom does not have a π -bonding to either D or A in the newly-developed D-O-A based organic phosphors. However, similar to anisole (see **Figure**), a p- π conjugation could happen between the n orbital from O and the π orbital from D or A, leading to the expected hybridization. As one can see (RTP-D2 in Figure 4d), the hole

distribution of S_1 is found to be extended from acridine (D) to O, and the hole distribution of T_1 is found to be extended from triazine (A) to O. Such a hybridization of n and π orbitals can favor the ISC process according to El-Sayed rule.

Figure. p- π conjugation in anisole.

Figure 4d

Most importantly, the hole migration from S_1 to T_1 means that holes will move from D to A. So the large dihedral angle (78.11° for RTP-D2 in Figure S22) may also allow enough orbital angular momentum change to compensate the spin angular momentum. In all, both the hybridization of n and π orbitals and the large dihedral angle do contribute to the ISC from S_1 to T_1 .

Figure S22. Dihedral angle between the acridine donor and the triazine acceptor for RTP-D1 (a) and RTP-D2 (b).

(2) **In RTP-D1:** The same situation is also suitable for RTP-D1 (Figure S25). Compared with S_1 , it is found that the hole of T_1 is extended from acridine and O to triazine, and the electron of T_1 is extended from triazine to O. During the hole and electron migrations from S_1 to T_1 , both the n orbital contribution and the large dihedral angle between D and A (64.50° for RTP-D1 in Figure S22) plays an important role on the orbital angular momentum change, which can compensate the spin angular momentum. Therefore, the spin flipping becomes more allowed from S_1 to T_1 than from S_1 to T_2 .

Figure S25. Energy level alignment together with the spin-orbit coupling matrix elements (a), and hole and electron distributions of S_1 and T_n below S_1 for RTP-D1. Compared with S_1 , it is found that the hole of T_1 is extended from acridine and O to triazine, and the electron of T_1 is extended from triazine to O. During the hole and electron migrations from S_1 to T_1 , both the n orbital contribution and the large dihedral angle between D and A (64.50° for RTP-D1 in Figure S22) plays an important role on the orbital angular momentum change, which can compensate the spin angular momentum. Therefore, the spin flipping becomes more allowed from S_1 to T_1 than from S_1 to T_2 .

(3) In our previous version, we are so sorry for the wrong SOCME data of RTP-D1. So we have corrected the mistakes, and included the related discussions in the revised manuscript. (Marked as yellow)

Q2: More strangely, why does the D-O-A structure produce quite unprecedentedly fast microsecond phosphorescence? The phosphorescence lifetime listed in Table 1 are very fast at 0.4 - 0.6 micro second about the same level of the organometallic phosphors, even though the delayed emission measurement was carried out at 100 micro seconds???. Where does the superefficient spin orbit coupling for this fast emission come from? This fast phosphorescence should not be sensitive to collisional quenching. Then how should explain the AIE? The same suspicion is on the oxygen quenching. Unlike purely organic phosphors having very slow triplet emission, fast phosphorescence emitters are not that sensitive to molecular triplet oxygen. However, the solution data in Figure S10 show almost complete quenching in in the presence of oxygen.

A: (1) **Microsecond phosphorescence:** On one hand, you know, there are several reports about organic phosphors showing microsecond phosphorescence, such as J. Phys. Chem. Lett. 2019, 10, 5983–5988; Adv. Mater. 2019, 31, 1904273; Chem. Mater. 2020, 32, 9, 4038–4044.

On the other hand, as you point out, besides the n orbital contribution from O, the large dihedral angle in the case of D-O-A is believed to be responsible for the fast microsecond phosphorescence at present. To demonstrate this point, we compare two similar molecules with different dihedral angle (**see Figure**). Based on RTP-D2, we design another molecule by fusing two phenyl rings around O. In this case, the dihedral angle between D and A is dropped to 0°. Although the n orbital of O

contributes to the hole-electron distributions of S_1 , T_1 and T_2 , they seem to match well with each other, leading to negligible SOCME values. This implies the unfavorable ISC and subsequent phosphorescence, quite different from RTP-D2 with a dihedral angle of 78.11° . So the fast microsecond phosphorescence in D-O-A may be tentatively attributed to the large dihedral angle between D and A in the D-O-A based organic phosphors. Further experiments should be performed to verify this hypothesis, but they are now beyond the aim of this work.

Figure. Comparison of simulation results for two molecules with different dihedral angle.

(2) **AIE:** It should be noted that the fast phosphorescence lifetime is detected in neat films for D-O-A based organic phosphors. Due to the weak signal, we could not

measure the phosphorescence lifetime in solution. As mentioned in the manuscript, AIE is related to the non-radiative decay (NR).

For organometallic phosphors (see **Figure**), all singlet excitons can be converted to triplet excitons due to the high ISC rate constant ($k_{ISC} > 10^8 \text{ S}^{-1}$). Meanwhile, NR is not a problem any longer because of the low NR rate constant ($k_{NR} < 10^5 \text{ S}^{-1}$). Therefore, they show strong phosphorescence even in solution.

Figure. Comparison between organometallic phosphors and D-O-A based organic phosphors in this work.

For D-O-A based organic phosphors, on one hand, due to the moderate k_{ISC} (10^7 S^{-1}), there is a singlet exciton residue, leading to weak fluorescence. On the other hand, due to the rotation around the C-O single bond (both D and A), NR ($k_{NR} > 10^6$

S^{-1}) plays an important role on the quenching of most of triplet excitons. Therefore, they show weak RTP in solution. In solid state, by contrast, such a NR can be effectively suppressed due to the rigid environment. Consequently, although the weak fluorescence remains unchanged, RTP is enhanced significantly. From this point of view, aggregation-induced RTP can be reasonably ascribed to the elimination of NR.

(3) **O₂ quenching:** As discussed above, D-O-A based organic phosphors display very weak RTP in solution. The weak RTP can be easily and completely quenched in the presence of O₂ (Figure S10). However, RTP becomes much more stronger in neat films, and shows a fast microsecond lifetime. In this case, part of RTP but not all is quenched by O₂ (Figure S9). The observation is similar to the behavior of organometallic phosphors in solution, indicative of less sensitivity to molecular triplet oxygen.

(4) To avoid the misunderstanding, the schematic illustration of AIE is provided as Figure S23 together with the related discussions. **(Marked as green)**

Q3: When it comes to the emission lifetime and the delay time, there is another example of confusion. Fig S18 shows the emissions at the 100 micro second delay time. At 300K, the emission at about 500nm is still observed. However, in Figure S17 the emission lifetime at 300 looks to be 5 - 10 microsecond which shouldn't be detected at 100 microsecond delay time.

A: (1) Thanks for your doubt. This is just a technical issue about the measurements of transient PL and steady-state PL.

(2) It should be noted that in the transient PL detected at about 500 nm, the PL intensity is found to be reduced to a very weak level at 100 microsecond delay time, but not decay to zero. Therefore, such a weak signal can still be detected after several magnifications in the steady-state PL.

First, TCSPC (Time-Correlated Single Photon Counting) is used in the transient PL, where only a single photon is detected to measure its decay behavior. However, in the steady-state PL for phosphorescence, pulsed xenon lamp is used as the excitation source, and a large number of photons other than a single photon are detected to measure their whole decay behavior.

Second, taking a certain point of wavelength as an example (see **Figure**). When a pulse excitation is imposed on the sample, in a steady-state mode, a large number of photons are generated. After a delay time (t_d) to eliminate the fluorescence signal, phosphorescence signal is then collected in a set time range (t_g). This integral phosphorescence signal can be further amplified by multiplying the pulse times (namely flash count) in a detected cycle. Based on the same principle, phosphorescence signals from all wavelengths are needed to be collected so as to provide the final phosphorescence spectrum.

Third, there are two monochromators including excitation and emission in the steady-state mode. Hence signal amplification can be achieved by increasing the Ex. Slit to generate more photons and Em. Slit to collect more emission.

With these magnifications, consequently, the weak signal in the transient PL can be detected in the steady-state PL for phosphorescence.

Figure. Tunable parameters in the steady-state PL for phosphorescence.

Q4: Another question is for the difference in the emission profiles of the solution and the neat film (for example, Figure S9 and S10). While the solution shows fluorescence and phosphorescence-like emission, the neat films emit mostly the phosphorescence peak at around 500nm. What boost so efficiently the intersystem crossing from the single to triplet in the neat film that is missing in the solution?

A: (1) Thanks for your doubt. You know, the emission profile of the solution is quite different from that of the neat film. This does not mean that efficient ISC is missing in the solution.

(2) Indeed, the different emission profiles are related to the AIE behavior. As discussed above (see Q2), RTP is weak in the solution due to the severe NR, whose intensity is comparable to the weak fluorescence. Therefore, an obvious dual emission

from fluorescence and RTP is observed in the solution without O₂.

Ongoing from solution to neat film, RTP is enhanced greatly due to the suppressed NR. In this case, RTP dominates the whole PL in neat film. However, the weak fluorescence is also involved, which can be differentiated from RTP via a Bigaussian fitting of the PL in neat film (Figure S15 and S20).

Q5: Figure 2. should show the PL spectra of the solutions in the panel (a). The PL spectra of the neat films are already in the panel (b).

A: (1) You know, the PL spectra (in the panel a) and phosphorescence spectra (in the panel b) are different. The PL spectra are measured without a delay, which include both fluorescence and RTP. However, the phosphorescence spectra are measured with a delay to collect only RTP.

(2) As suggested, the PL spectra in the solution without O₂ are added in Figure 2 to clearly demonstrate the difference between solution and neat film. (**Marked as sky-blue**)

Q6: Fig S16. The emission lambda max in THF is at around 550 nm???. If the molecular motion causes the quenching of the emission there shouldn't be emission lambda max change.

Figure 3 (a) should show whole spectrum from the excitation wavelength. What is the excitation wavelength? Why does the emission color (emission lambda max) gradually change? Shouldn't only the emission intensity increase?

A: (1) Thanks for your carefulness. You are right. With the increasing water fraction, the emission maxima are blue-shifted together with the enhanced intensity (Figure 3 and S16).

(2) As discussed above (see **Q2 and Q4**), the PL in solution with O₂ is corresponding to the fluorescence, because the weak RTP is completely quenched by O₂. Due to the charge transfer (CT) nature, fluorescence is sensitive to the solvent polarity. To demonstrate this point, the PL spectra are measured at different organic solvents for RTP-D1 and RTP-D2. It is found that the maximum emission appears at about 575 nm in THF (Figure S24).

However, in aggregates similar to neat films, the corresponding PL is mainly from RTP peaked at about 500 nm. Accordingly, the PL spectra are blue-shifted with the increasing water fraction.

Figure S24. PL spectra measured in different organic solvents in the presence of O₂ for RTP-D1 (a) and RTP-D2 (b). As for their AIE behaviors (Figure 3 and S16), the PL spectra also show a hypsochromic shift with the increasing water fraction.

As discussed above, the PL in solution with O₂ is corresponding to the fluorescence, because the weak RTP is completely quenched by O₂. Due to the charge transfer (CT) nature, fluorescence is sensitive to the solvent polarity. So the emission maxima of RTP-D1 and RTP-D2 appear at about 575 nm in THF. However, in aggregates, the corresponding PL is dominantly from RTP peaked at about 500 nm. Therefore, the PL spectra are found to be blue-shifted with the increasing water fraction.

(3) The related explanation about the hypsochromatic shift is given in Figure S24.

Also, the excitation wavelength is added in Figure 3 as suggested. **(Marked as pink)**

Q7: Hoped to see a better balance in reference because two seminal papers having high citation numbers and demonstrating the very early stage of rational organic phosphor design should have been cited (Nat. Mater. 2009, 8, 747 & Nat. Chem. 2011, 3, 205)

A: We are so sorry for the missing of these two seminal papers. They have been cited as Ref. 10 and 12 in the revised manuscript. **(Marked as grey)**

Reviewers' Comments:

Reviewer #2:

Remarks to the Author:

The main issue with this paper is the origin of the long-lived emission observed in compounds RTP-D1 and RTP-D2, regarding whether it is delayed fluorescence or phosphorescence.

The authors have demonstrated that in both solution and solid film an emission band peaking around 500 nm is sufficiently red-shifted and is also too long-lived to be assigned to the normal fluorescence emission in these compounds.

The long-lived emission band appears in solution peaking around 500 nm, relative to the normal fluorescence band that peaks around 450 nm. The long-lived, red-shifted, emission is totally quenched by oxygen, which indicates it is originated by triplet excited states. In solid state two distinct bands are found to form the broad luminescence spectrum. The red-shifted band observed in solid state matches well the emission spectrum of the long-lived emission observed in solution and is equally long-lived. Moreover, the intensity of the long-lived emission (collected after a delay time of 3 microseconds) increases with decreasing temperature which indicates this emission is due to phosphorescence.

The reviewers made several relevant questions which, in my opinion, the authors responded satisfactorily.

In regard to the EL vs PL mechanism, i.e. whether the EL is due to phosphorescence or not, the authors argue with the fact that they could demonstrate the PL is due to phosphorescence and that the EL matches the PL emission, which is frequently accepted as sufficient evidence to ensure PL and EL have the same origin. I think the authors could also measure the EL lifetime to give additional evidence.

The fact the long-lived emission decays entirely in less than 5 microseconds, whereas the gated delayed emission is measured with 100 microseconds delay time can also be explained by the technicality that the decay was measured with constant integration time, which makes it difficult to collect sufficient photons at longer times. However, the gated emission with longer integration time can collect emission which seem not being present in the decay.

In summary, in my opinion, the data given in this work indicates within reasonable confidence that the PL and EL can be ascribed to RTP. Therefore, since there are very few cases in the literature of EL based RTP devices, the manuscript should be accepted for publication in its current form.

Point-by-point response to the reviewer #2

Q1: The main issue with this paper is the origin of the long-lived emission observed in compounds RTP-D1 and RTP-D2, regarding whether it is delayed fluorescence or phosphorescence.

The authors have demonstrated that in both solution and solid film an emission band peaking around 500 nm is sufficiently red-shifted and is also too long-lived to be assigned to the normal fluorescence emission in these compounds.

The long-lived emission band appears in solution peaking around 500 nm, relative to the normal fluorescence band that peaks around 450 nm. The long-lived, red-shifted, emission is totally quenched by oxygen, which indicates it is originated by triplet excited states. In solid state two distinct bands are found to form the broad luminescence spectrum. The red-shifted band observed in solid state matches well the emission spectrum of the long-lived emission observed in solution and is equally long-lived. Moreover, the intensity of the long-lived emission (collected after a delay time of 3 microseconds) increases with decreasing temperature which indicates this emission is due to phosphorescence.

The reviewers made several relevant questions which, in my opinion, the authors responded satisfactorily.

In regard to the EL vs PL mechanism, i.e. whether the EL is due to phosphorescence or not, the authors argue with the fact that they could demonstrate the PL is due to phosphorescence and that the EL matches the PL emission, which is frequently accepted as sufficient evidence to ensure PL and EL have the same origin. I think the

authors could also measure the EL lifetime to give additional evidence.

The fact the long-lived emission decays entirely in less than 5 microseconds, whereas the gated delayed emission is measured with 100 microseconds delay time can also be explained by the technicality that the decay was measured with constant integration time, which makes it difficult to collect sufficient photons at longer times. However, the gated emission with longer integration time can collect emission which seem not being present in the decay.

In summary, in my opinion, the data given in this work indicates within reasonable confidence that the PL and EL can be ascribed to RTP. Therefore, since there are very few cases in the literature of EL based RTP devices, the manuscript should be accepted for publication in its current form.

A: (1) You know, as a new field, there are lots of issues to be addressed, especially the mechanism. And the good suggestions have helped us improve the quality of our manuscript greatly. So many thanks for your acceptance of our second revisions.

(2) According to your suggestion, the transient EL spectrum has been measured and provided as Supplementary Figure 28a.

Supplementary Figure 28. Transient EL spectrum driven at a 7 V pulse (a) and temperature dependence of the EL spectra (b) for RTP-D2 based non-doped device. After switching off the electrical pulse, an obvious delay is observed in the transient EL, well consistent with the transient PL.